# Optimizing Safety and Efficacy of Intravenous Vancomycin Therapy in Orthopedic Inpatients Through a Standardized Dosing Protocol: A Pre-Post Cohort Study

**DOI:** 10.3390/antibiotics14080775

**Published:** 2025-07-31

**Authors:** Moritz Diers, Juliane Beschauner, Maria Felsberg, Alexander Zeh, Karl-Stefan Delank, Natalia Gutteck, Felix Werneburg

**Affiliations:** Department of Orthopaedic, and Trauma Surgery, Martin-Luther-University Halle-Wittenberg, 06120 Halle, Germany; moritz.diers@uk-halle.de (M.D.);

**Keywords:** vancomycin, orthopedic infections, antibiotic stewardship, acute kidney injury

## Abstract

**Background**: Intravenous vancomycin remains a key agent in the treatment of complex orthopedic infections, particularly those involving methicillin-resistant *Staphylococcus aureus* (MRSA). However, its use is associated with significant risks, most notably nephrotoxicity. Despite guideline recommendations, standardized dosing and monitoring protocols are often absent in orthopedic settings, leading to inconsistent therapeutic drug exposure and preventable adverse events. This study evaluated the clinical impact of implementing a structured standard operating procedure (SOP) for intravenous vancomycin therapy in orthopedic inpatients. **Methods**: We conducted a single-center, pre-post cohort study at a university orthopedic department. The intervention consisted of a standard operating procedure (SOP) for intravenous vancomycin therapy, which mandated weight-based loading doses, renal function-adjusted maintenance dosing, trough level monitoring, and defined dose adjustments. Patients treated before SOP implementation (*n* = 58) formed the control group; those treated under the SOP (*n* = 56) were prospectively included. The primary outcome was the incidence of vancomycin-associated acute kidney injury (VA-AKI) defined by KDIGO Stage 1 criteria. Secondary outcomes included therapeutic trough level attainment and infusion-related or ototoxic adverse events. **Results:** All patients in the post-SOP group received a loading dose (100% vs. 31% pre-SOP, *p* < 0.001). The range of measured vancomycin trough levels narrowed substantially after SOP implementation (7.1–36.2 mg/L vs. 4.0–80.0 mg/L). The proportion of patients reaching therapeutic trough levels increased, although this was not statistically significant. Most notably, VA-AKI occurred in 17.2% of patients in the control group, but in none of the patients after SOP implementation (0%, *p* = 0.0013). No cases of ototoxicity were observed in either group. Infusion-related reactions decreased after the implementation of the SOP, though not significantly. **Conclusions**: The introduction of a structured vancomycin protocol significantly reduced adverse drug events and improved dosing control in orthopedic inpatients. Incorporating such protocols into routine practice represents a feasible and effective strategy to strengthen antibiotic stewardship and clinical quality in surgical disciplines.

## 1. Introduction

Vancomycin remains a cornerstone in the treatment of serious Gram-positive infections, particularly those caused by methicillin-resistant *Staphylococcus aureus* (MRSA). In an orthopedic surgery, intravenous vancomycin plays a central role in the management of periprosthetic joint infections, osteomyelitis, and implant-associated complications—clinical scenarios that frequently require long-term antibiotic therapy in patients with increased vulnerability due to age and comorbidities [1].

Postoperative infections remain among the most feared complications in orthopedic surgery due to their significant morbidity and potential long-term impairment. A large-scale analysis of 359,268 inpatient orthopedic surgical encounters in the United States reported an overall incidence of S. aureus infections of 1.13%, with surgical site infections (SSIs) accounting for 0.68% and bloodstream infections (BSIs) for 0.28%. Among culture-confirmed SSIs, S. aureus was the most frequently isolated pathogen (48%) [2]. MRSA continues to represent a clinically relevant multidrug-resistant organism in orthopedic infections. According to the U.S. Centers for Disease Control and Prevention (CDC), approximately 43% of hospital-associated S. aureus infections are attributable to MRSA, with even higher rates reported in surgical disciplines such as orthopedics. European surveillance data from the ECDC confirm this trend, with MRSA proportions ranging from below 5% in Northern Europe to over 25% in parts of Southern and Eastern Europe [3]. In orthopedic procedures, MRSA is a major pathogen in periprosthetic joint infections and postoperative wound complications—conditions that often require extended intravenous antibiotic treatment. In addition to MRSA, coagulase-negative staphylococci (CoNS), particularly *Staphylococcus epidermidis*, are increasingly recognized as major pathogens in chronic implant-associated infections. While often considered contaminants in other clinical contexts, these organisms are highly relevant in orthopedic surgery due to their ability to form biofilms on prosthetic materials and hardware. CoNS are frequently implicated in periprosthetic joint infections and low-grade infections, where they pose significant diagnostic and therapeutic challenges.

While clinically indispensable, systemic vancomycin therapy is associated with several well-documented adverse effects. Among these, vancomycin-associated acute kidney injury (VA-AKI) is the most common and potentially serious, with reported incidence rates ranging from 5% to over 30%, depending on patient population and dosing practices [4]. Additional adverse events include infusion-related reactions such as the vancomycin flush syndrome (Red Man Syndrome) [5] and, less commonly, ototoxicity, particularly in elderly patients or those with renal impairment [6]. These risks are closely linked to both peak serum concentrations and cumulative drug exposure, highlighting the importance of structured and individualized vancomycin administration [7].

Therapeutic success also hinges on achieving and maintaining pharmacodynamic targets. Current consensus guidelines recommend an area under the concentration-time curve to minimum inhibitory concentration (AUC/MIC) ratio of 400–600 as the optimal therapeutic range [8]. Attaining this target has been shown to enhance bactericidal activity while significantly reducing the risk of nephrotoxicity, particularly when compared to traditional trough-based dosing [9,10]. The key components of contemporary vancomycin management include early therapeutic drug monitoring (TDM), weight-based loading dose administration, and renal function-adjusted maintenance dosing.

Despite these evidence-based recommendations, their consistent application in orthopedic settings remains limited. In contrast to internal medicine and intensive care units, where standardized dosing protocols and pharmacy-led stewardship programs are increasingly implemented [11], orthopedic departments often lack structured frameworks for vancomycin therapy. As a result, dosing and monitoring decisions are frequently left to individual clinical judgment, without standardized guidance on loading doses, TDM, or dose adjustments [9]. This contributes to significant variability in drug exposure, the timing of level measurements, and the adequacy of therapeutic modifications. Such inconsistencies may lead to subtherapeutic concentrations and reduced efficacy, or drug overexposure with increased risk of preventable toxicity [12,13].

To address these challenges, a structured standard operating procedure (SOP) for intravenous vancomycin therapy was implemented at the Department of Orthopedic Surgery of our university hospital. The SOP aimed to enhance the therapeutic precision of vancomycin therapy while simultaneously reducing the risk of avoidable adverse effects. The protocol incorporated several key measures to achieve both objectives: timely administration of a weight-based loading dose, renal function–adjusted and regularly scheduled monitoring of vancomycin trough levels (TDM), clearly defined thresholds for dose adjustments, and standardized infusion durations. This study evaluates the clinical impact of the SOP using a pre-post cohort design. Patients treated prior to SOP implementation formed a retrospectively analyzed control group, while patients managed under the SOP were prospectively monitored. We hypothesized that this structured approach to vancomycin therapy would improve pharmacological target attainment and reduce the incidence of adverse drug events in orthopedic inpatients.

## 2. Materials and Methods

### 2.1. Study Design and Setting

This study was designed as a single-center, non-randomized interventional study employing a pre-post design to evaluate the clinical impact of a standardized vancomycin dosing and monitoring protocol. All patients were treated at the Department of Orthopedic Surgery at our university hospital. Data were collected retrospectively for the control group (pre-SOP) and prospectively for the intervention group (post-SOP) between 2023 and 2024.

### 2.2. Patient Selection

Patients were eligible for inclusion if they received intravenous vancomycin therapy during their inpatient stay at the orthopedic department, either as part of empiric combination treatment (typically ampicillin/sulbactam plus vancomycin) for presumed or culture-negative infections, or as targeted monotherapy with vancomycin in microbiologically confirmed cases. Patients were excluded if vancomycin was administered as a single dose only, if the duration of therapy was less than 72 h, or if severe renal dysfunction was present at baseline, defined as chronic kidney disease stage 4 or higher according to KDIGO criteria (eGFR < 30 mL/min/1.73 m^2^) or the need for dialysis. Additional exclusion criteria included the concomitant use of nephrotoxic agents during vancomycin therapy (such as aminoglycosides or NSAIDs), incomplete documentation of vancomycin trough levels or renal function parameters, and antimicrobial therapy initiated or continued outside the orthopedic department. In total, 156 patients were screened for eligibility, of whom 42 were excluded based on the above criteria. This included 9 patients who had received concomitant aminoglycoside therapy. After applying these criteria, a total of 114 patients were included in the final analysis: 58 in the control group (prior to SOP implementation) and 56 in the intervention group (after SOP implementation).

### 2.3. Intervention: Standard Operating Procedure

In March 2024, a structured standard operating procedure (SOP) for intravenous vancomycin therapy was implemented at the Department of Orthopedic Surgery at our university hospital. No additional staffing or external resources were required. Implementation steps included (1) distribution of the written protocol via internal communication channels, (2) brief in-service training for physicians and nursing staff, and (3) integration of key SOP components into daily ward rounds. These measures were embedded into existing clinical workflows, making the SOP low-threshold and resource-conscious. The SOP was developed to enhance both treatment safety and therapeutic precision by introducing standardized dosing practices and therapeutic drug monitoring (TDM). It was applied exclusively within the orthopedic service and followed institutional quality standards while aligning with internationally accepted guidelines for vancomycin use in serious MRSA infections, including the revised consensus recommendations issued by ASHP, IDSA, PIDS, and SIDP [9].

The key elements of the SOP included the mandatory administration of a weight-based loading dose (20–25 mg/kg), renal function–adjusted maintenance dosing, a predefined target trough range of 15–20 mg/L, and clearly specified timing for TDM based on glomerular filtration rate (GFR). Trough levels were measured at steady state, defined as within 1 h prior to the respective dose, depending on renal function:-before the 4th dose in patients with GFR ≥ 40 mL/min/1.73 m^2^,-before the 3rd dose in patients with GFR 20–39 mL/min/1.73 m^2^,-24 h after the loading dose in patients with GFR < 20 mL/min/1.73 m^2^ or receiving dialysis.

Dose adjustments were made using a structured matrix (Table 1) that integrated both trough level and renal function, with re-dosing intervals modified accordingly. In patients with impaired renal function, re-dosing was guided by repeated TDM, with vancomycin administered only when serum levels dropped below 20 mg/L.

In addition, to prevent infusion-related reactions such as vancomycin flush syndrome, we standardized infusion durations based on total dose (Table 2). Doses < 1 g were infused over at least 60 min; 1.1–1.5 g over 90 min; 1.6–1.9 g over 120 min; and doses > 2 g at a maximum rate of 1 g per 60 min.

### 2.4. Data Collection

For all patients, the following data were collected: age, sex, presence of a loading dose (yes/no), vancomycin trough concentrations, baseline renal function (estimated GFR), and the occurrence of adverse events, including nephrotoxicity, ototoxicity, and vancomycin flush syndrome. For patients in the intervention group, additional data were recorded: indication for vancomycin use, treatment rationale (empiric vs. targeted), identified pathogen (if applicable), body weight, exact dosing (loading and maintenance), start date of therapy, timing of the first trough level measurement, number and values of all measured trough levels, and total duration of vancomycin therapy in days.

### 2.5. Outcome Measures

The primary outcome was the incidence of vancomycin-associated nephrotoxicity (VA-AKI), defined in accordance with KDIGO Stage 1 criteria as an increase in serum creatinine of ≥0.3 mg/dL or ≥1.5 times baseline, without consideration of urine output. A key secondary outcome was therapeutic target attainment, defined as a vancomycin trough level between 15 and 20 mg/L at the first measurement. Other outcomes included the frequency of adverse drug reactions such as ototoxicity or vancomycin flush syndrome, based on clinical documentation.

### 2.6. Statistical Analysis

All data were collected and pseudonymized using Microsoft Excel and subsequently analyzed using SPSS Statistics (IBM SPSS Statistics for Windows, Version 27.0. Armonk, NY, USA: IBM Corp.). Descriptive statistics were calculated for both the control and intervention groups. Continuous variables were summarized as means with standard deviations (SD) or medians with interquartile ranges (IQR), as appropriate. Categorical variables were expressed as absolute numbers and percentages.

Group comparisons for continuous variables were conducted using independent samples *t*-tests. Welch’s correction was applied in cases of unequal variances between groups. For categorical variables, chi-square tests or Fisher’s exact tests were used as appropriate. A *p*-value of <0.05 was considered statistically significant for all comparisons.

### 2.7. Ethical Approval

For this retrospective analysis of anonymized data, the requirement for formal ethical approval was waived in accordance with institutional policy. A declaration of no objection was issued by the ethics committee of the Medical Faculty at Martin Luther University Halle-Wittenberg.

## 3. Results

### 3.1. Patient Characteristics

A total of 114 patients were included in this study: 58 in the control group (prior to SOP implementation) and 56 in the intervention group (post-SOP). An overview of baseline demographic and clinical characteristics of both study groups is provided in Table 3.

### 3.2. Loading Dose, Initial Trough Level Concentrations, and Trough Level Monitoring

In the control group, only 18 of 58 patients (31%) received a vancomycin loading dose. In contrast, all 56 patients (100.0%) in the intervention group were administered a weight-based loading dose in accordance with the standardized protocol.

At the time of the first trough level measurement, the mean serum vancomycin concentration was 15.1 mg/L (SD ± 10.6) in the control group and 19.0 mg/L (SD ± 8.0) in the intervention group. Statistical comparison using Welch’s *t*-test yielded a *p*-value of 0.079.

The mean vancomycin trough level across all measurements was 18.1 mg/L (SD ± 9.9; range 4.0–80.0 mg/L) in the control group and 18.6 mg/L (SD ± 6.0; range 7.1–36.2 mg/L) in the intervention group. Statistical comparison using Welch’s *t*-test yielded a *p*-value of 0.555, indicating no statistically significant difference in the average trough concentrations between groups.

### 3.3. Adverse Events

Among the 58 patients in the control group, vancomycin-associated acute kidney injury (VA-AKI), defined in accordance with KDIGO Stage 1 criteria, occurred in 10 cases. In the intervention group, no cases of VA-AKI were observed. Statistical comparison using Fisher’s exact test yielded a statistically significant result (*p* = 0.0013). The distribution of VA-AKI events in both groups is illustrated in Figure 1.

Vancomycin flush syndrome was reported in 5.2% of patients in the control group (*n* = 3), while no cases occurred in the intervention group (0.0%) (*p* = 0.243, Fisher’s exact test).

No instances of ototoxicity were documented in either cohort.

## 4. Discussion

### 4.1. Principal Findings and Clinical Relevance

In this single-center pre/post cohort study, implementing a standardized vancomycin dosing SOP in orthopedic patients led to markedly improved dosing metrics and patient safety outcomes. All patients in the post-implementation group received a weight-based loading dose, compared to inconsistent use previously, ensuring rapid attainment of therapeutic drug levels. Correspondingly, vancomycin trough concentrations became much more tightly controlled, and the observed range narrowed from 4.0–80.0 mg/L before to 7.1–36.2 mg/L after SOP introduction. This indicates fewer extreme sub-therapeutic or supratherapeutic values, reflecting more precise dosing. A greater proportion of trough levels fell within the recommended 15–20 mg/L target range in the post-SOP period. Although this increase did not reach statistical significance, it demonstrated a positive trend toward improved target attainment. Most importantly, vancomycin-associated acute kidney injury (VA-AKI) was virtually eliminated after SOP implementation. VA-AKI incidence dropped from 17.2% of patients in the pre-SOP cohort to 0% in the post-SOP cohort, a statistically significant reduction (*p* = 0.0013). This finding is clinically highly relevant, as nephrotoxicity is a well-known hazard of vancomycin therapy, with reported incidence in the literature ranging from about 1% up to 40% depending on patient populations and dosing intensity [14]. Our 17.2% baseline VA-AKI rate is consistent with the higher end of typical reports for aggressive trough-based regimens, whereas our SOP results strongly suggest that a structured vancomycin dosing protocol can improve patient safety and dosing effectiveness in orthopedic practice. By ensuring appropriate loading doses and maintenance dose adjustments, the SOP likely prevented both under-dosing (which risks treatment failure) and over-dosing (which increases toxicity). Thus, maintaining trough levels in a narrower, safer window through diligent protocolized dosing has immediate clinical payoff. Our findings align with emerging evidence that optimizing vancomycin exposure improves safety. For example, recent pharmacodynamic studies show that limiting vancomycin exposure (e.g., via AUC-guided dosing) can significantly reduce nephrotoxicity without compromising efficacy [9]. In our study, we achieved a similar goal—reducing undue vancomycin exposure—but through a pragmatic trough-based protocol tailored to our setting. The fact that no patient experienced nephrotoxic injury post-intervention is compelling for clinicians, as acute kidney injury can prolong hospitalization, delay surgical rehabilitation, and necessitate complex care (dialysis or drug changes). In summary, the SOP’s implementation led to safer vancomycin therapy and potentially more effective treatment of infections by keeping drug levels in the therapeutic sweet spot.

### 4.2. Antibiotic Stewardship and Quality Improvement in Orthopedics

Our study’s intervention can be viewed through the lens of antibiotic stewardship and quality improvement in a surgical discipline. Antibiotic stewardship principles were at the core of the SOP. We standardized dosing to use the right dose at the right time for each patient, thereby optimizing efficacy while minimizing toxicity. The substantial reduction in AKI exemplifies how stewardship is not only about combating resistance but also about improving patient outcomes by reducing antibiotic-related harm. Orthopedic surgery patients—especially those with serious infections like osteomyelitis or prosthetic joint infections—often require intensive antibiotic therapy, such as prolonged intravenous vancomycin. Ensuring these therapies are managed optimally is a quality-of-care issue. Our findings demonstrate that a collaboration between orthopedic teams, infectious disease specialists, and clinical pharmacists to implement best practices can measurably improve patient safety in surgical care. This mirrors reports from other surgical stewardship efforts; for instance, integrating an antimicrobial stewardship program (ASP) into elective orthopedic surgery has been associated with more appropriate antibiotic use and lower rates of postoperative complications, including AKI [15]. Likewise, adjustments in surgical antibiotic protocols elsewhere have led to reduced nephrotoxic risk—one study observed that changing the standard prophylactic regimen in orthopedic surgery (from a nephrotoxic combination to an alternative agent) significantly decreased post-operative AKI rates [16]. These examples underscore that orthopedic departments can benefit from stewardship-driven protocol changes, both in prophylaxis and therapeutic antibiotic use.

By achieving 100% uptake of loading doses and consistent therapeutic drug monitoring, our SOP ensured that no patient fell through the cracks of variable practice. This level of compliance is noteworthy. In comparison, a recent evaluation of vancomycin guideline implementation in a large hospital setting achieved around 84% compliance with recommended loading doses, yet found that only about ~30% of initial trough levels were in the target range [17]. In our orthopedic unit, the tighter distribution of trough levels suggests that standardization led to more reliable attainment of desired concentrations. This reliability is crucial in surgical patients: under-dosing vancomycin could lead to persistent infection, whereas overdosing could cause toxicity—both scenarios jeopardize surgical outcomes. Embracing stewardship in orthopedics—through protocols like ours—not only helps combat antimicrobial resistance but also directly improves perioperative care by reducing adverse drug events.

### 4.3. Comparison with Current Vancomycin Dosing Strategies

It is important to interpret our results in the context of evolving vancomycin dosing strategies. At the time of our protocol’s use, trough concentration monitoring was the standard in our center for guiding therapy. We chose the target trough range of 15–20 mg/L, consistent with consensus guidelines (2009 IDSA guidelines) for serious MRSA infections. Our SOP’s success in narrowing trough variability and preventing toxicity demonstrates that significant improvements are achievable even within a trough-based dosing framework. However, contemporary practice is shifting toward area-under-the-curve (AUC)-guided dosing for vancomycin. The latest consensus guidelines (2020) explicitly recommend AUC/MIC targets (typically 400–600 mg·h/L) rather than high trough levels, citing evidence that trough-only targeting of 15–20 mg/L is associated with excess nephrotoxicity [9]. AUC-guided dosing aims to maintain efficacy while lowering the incidence of AKI by avoiding unnecessarily high vancomycin exposure. Indeed, multiple studies have shown that switching from trough-based to AUC-based monitoring can cut nephrotoxic risk—for example, one meta-analysis found significantly lower AKI rates (on the order of ~1–2% incidence) with AUC-guided strategies compared to traditional trough approaches [8]. Our study did not incorporate AUC monitoring or Bayesian dosing software, and this is a notable limitation (discussed further below). Nonetheless, our approach—emphasizing loading doses and vigilant trough monitoring/adjustment—aligns with the overarching goal of minimizing vancomycin toxicity. By reducing the occurrence of extreme trough values, we essentially approximated a safer dosing profile. In centers where AUC-guided pharmacy services are not yet available, our experience suggests that a well-designed trough-based protocol can be a major step forward in improving vancomycin safety.

It is also worth noting that our zero AKI rate in the post-SOP group compares favorably with most published experiences, even those using advanced dosing methods [18], which report considerably lower-than-average rates, but still not zero. The complete absence of nephrotoxicity in our cohort may reflect the high level of adherence to the protocol. All patients were managed closely by a dedicated team, and any early signs of kidney function change likely prompted immediate dose adjustments or supportive measures. While AUC-based dosing is ideal, our findings highlight that diligent stewardship within existing dosing paradigms can nearly eliminate a major adverse effect. This is encouraging for hospitals that are in the process of transitioning to AUC-guided monitoring; interim measures like standardizing trough-based dosing can yield significant benefits. Looking forward, combining our SOP with AUC-guided monitoring (when feasible) could be an attractive strategy to further enhance therapeutic precision. For example, adding real-time AUC calculations to our protocol might improve target attainment beyond the observed “trend” and ensure optimal exposure for efficacy. Some pilot studies have reported success with this hybrid approach—implementing AUC-guided protocols with multidisciplinary stewardship support drove protocol adherence above 90% and even reduced 30-day mortality [18]. Thus, continuous refinement of vancomycin dosing strategies, whether through improved protocols or new monitoring tools, remains a pertinent consideration for sustaining the balance between efficacy and safety.

### 4.4. Alternative Treatment Options and Implementation Considerations

While vancomycin remains the first-line agent for serious Gram-positive infections in orthopedic settings, it is important to acknowledge that alternative antibiotics such as daptomycin are increasingly being used in clinical practice [19]. Daptomycin offers several practical advantages, including rapid bactericidal activity, predictable pharmacokinetics, and the absence of complex therapeutic drug monitoring requirements. Unlike vancomycin, daptomycin does not require serum trough or AUC level measurement, and it carries a substantially lower risk of nephrotoxicity [20]. This makes it particularly attractive in patients with pre-existing renal impairment or where regular pharmacokinetic monitoring is logistically difficult. Moreover, daptomycin has demonstrated efficacy in various orthopedic infections, including osteomyelitis and prosthetic joint infections, especially when used in combination with rifampicin for biofilm-associated infections [21].

However, daptomycin’s higher acquisition cost compared to generic vancomycin remains a limiting factor in many institutions, particularly when used for extended treatment durations [22]. In this context, the implementation of a vancomycin SOP represents a resource-conscious strategy to improve safety and efficacy without significantly increasing drug-related costs. Our findings suggest that even within a conventional vancomycin-based regimen, structured dosing protocols can lead to excellent safety outcomes, potentially mitigating the need for more expensive alternatives in certain patients.

That said, the successful implementation of any SOP depends on local infrastructure and resource availability. Although no additional personnel or software were required in our setting, this may not be the case elsewhere. In hospitals with limited pharmacy support, laboratory turnaround times, or staffing flexibility, the adoption of vancomycin protocols—especially those requiring close monitoring—may face logistical hurdles. Institutions should, therefore, assess the feasibility of such SOPs within their own operational framework and consider simplified or alternative approaches when appropriate. Future investigations might also evaluate whether integrating agents like daptomycin into SOP-driven treatment algorithms could further streamline care, particularly in high-risk or difficult-to-monitor patient populations.

### 4.5. Limitations

This study has several limitations that should be considered when interpreting the findings. First, the pre-post design without a contemporaneous control group introduces susceptibility to temporal confounding. Improvements observed after SOP implementation—particularly the reduction in vancomycin-associated nephrotoxicity—may partially reflect heightened clinical awareness, evolving institutional practices, or unmeasured improvements in supportive care over time. While the magnitude of effect and mechanistic plausibility support a causal role of the protocol, a randomized controlled trial would be required to definitively exclude alternative explanations.

Second, the study was conducted in a single academic orthopedic department. Although this ensures consistency in clinical practice, it may limit the generalizability of the findings. Patient populations, infrastructure, and adherence to protocolized care can vary across hospitals. Therefore, the effectiveness of a similar SOP may differ in community settings or other surgical specialties. External validation in multicenter or real-world cohorts would be valuable.

Third, although the sample size was adequate to detect a significant difference in the incidence of VA-AKI, it may have been insufficient to identify smaller but clinically relevant effects, such as differences in the proportion of patients achieving pharmacodynamic target trough levels. Some trends, such as improved target-range attainment in the post-SOP group, did not reach statistical significance and may become clearer in larger studies.

Fourth, the study did not include pharmacokinetic data on area under the curve (AUC) or minimum inhibitory concentrations (MICs), relying on trough levels as a surrogate for vancomycin exposure. Given that AUC-guided dosing is now considered the pharmacodynamic gold standard for balancing efficacy and nephrotoxicity risk, the absence of AUC monitoring is a noteworthy limitation. Without these data, it remains unclear whether the observed improvements in safety were fully aligned with optimal drug exposure.

Finally, we focused primarily on dosing parameters and safety outcomes such as nephrotoxicity. Clinical endpoints such as microbiological eradication, duration of bacteremia, surgical site reinfection, and overall treatment success were not assessed. While improved drug level control and avoidance of toxicity are important surrogate markers, future studies should evaluate whether protocolized dosing translates into better infection-related and functional outcomes in orthopedic patients.

## 5. Conclusions

The implementation of a standardized vancomycin dosing protocol in an orthopedic inpatient setting was associated with a significant reduction in nephrotoxicity and improved dosing consistency. By ensuring universal application of weight-based loading doses, structured therapeutic drug monitoring, and renal function-adjusted maintenance dosing, the protocol enhanced patient safety without compromising therapeutic efficacy. These findings highlight the clinical value of targeted antibiotic stewardship in surgical specialties and support the adoption of protocolized vancomycin management as a pragmatic quality improvement strategy in orthopedic care. Future research should assess the impact on infection-related outcomes and explore integration with AUC-guided monitoring frameworks.

## Figures and Tables

**Figure 1 antibiotics-14-00775-f001:**
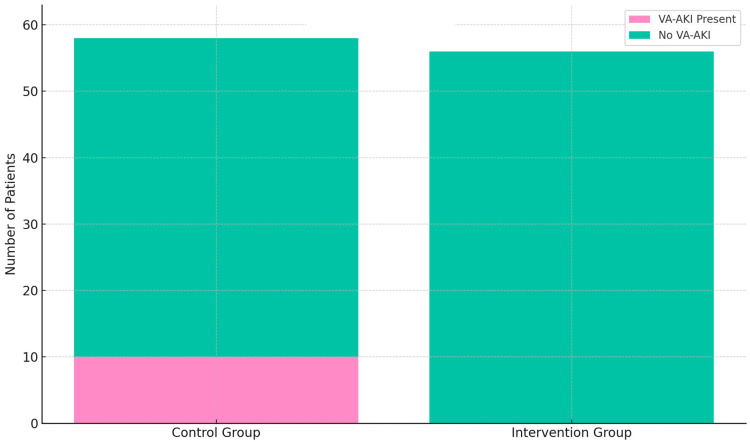
Incidence of vancomycin-associated acute kidney injury (VA-AKI) in the control group (*n* = 58) and intervention group (*n* = 56). VA-AKI occurred in 10 patients (17.2%) in the control group, whereas no cases were observed in the intervention group. Statistical comparison using Fisher’s exact test yielded a *p*-value of 0.0013.

**Table 1 antibiotics-14-00775-t001:** Vancomycin dosing adjustments based on trough level and renal function.

Trough Level (mg/L)	GFR (mL/min/1.73 m^2^) > 90	GFR 60–90 (mL/min/1.73 m^2^)	GFR 40–59 (mL/min/1.73 m^2^)	GFR 20–39 (mL/min/1.73 m^2^)	GFR < 20 (mL/min/1.73 m^2^)
<10	4 × 1 g	2 × 1.5 g	2 × 1 g	2 × 750 mg	TDM every 48 h; dose 1 g when through level < 20 mg/L
10–14.9	3 × 1.25 g	2 × 1.25 g	2 × 1 g	2 × 750 mg	
15–20	No dose adjustment required				
20.1–25	2 × 1.25 g	2 × 750 mg	2 × 500 mg	1 × 750 mg	
25.1–30	2 × 1 g	2 × 750 mg	2 × 500 mg	1 × 750 mg	
>30	Hold 24 h, recheck level				

Dosing recommendations were adapted according to estimated glomerular filtration rate (eGFR, mL/min) and measured vancomycin trough concentrations. For patients with severe renal impairment (eGFR < 20 mL/min), re-dosing was guided by repeated therapeutic drug monitoring every 48 h.

**Table 2 antibiotics-14-00775-t002:** Minimum infusion durations by vancomycin dose. Doses exceeding 2 g were administered at a maximum rate of 1 g per 60 min.

Vancomycin Dose (g)	Minimum Infusion Duration (min)
<1	60
1.1–1.5	90
1.6–1.9	120
>2	Approx. 1 g per 60 min

**Table 3 antibiotics-14-00775-t003:** Baseline demographic and clinical characteristics of orthopedic inpatients receiving intravenous vancomycin, stratified by study group (pre-SOP vs. post-SOP).

Characteristic	Pre-SOP (Control Group)	Post-SOP (Intervention Group)
**Demographics**		
Number of patients (*n*)	58	56
Mean age (SD), years	67.1 (9.9)	64.7 (11.1)
Male sex (*n*)	28	27
**Clinical Indication for Vancomycin Therapy (*n*)**		
Periprosthetic joint infection	33	34
Septic arthritis	12	12
Soft tissue infection	8	7
Spondylodiscitis	5	3
**Type of Vancomycin Therapy (*n*)**		
Empiric therapy	36	37
Targeted therapy	22	19
**Pathogens Identified (Targeted Therapy Only) (*n*)**		
*Staphylococcus epidermidis*	12	13
*Staphylococcus aureus*	7	5
*Staphylococcus capitis*	3	1
**Duration of Vancomycin Therapy**		
Mean (SD), days	8.6 (4.7)	9.1 (4.4)

## Data Availability

The data presented in this study are available on request from the corresponding author. The data are not publicly available due to privacy and institutional restrictions.

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
