# Peer review of "Optimizing Safety and Efficacy of Intravenous Vancomycin Therapy in Orthopedic Inpatients Through a Standardized Dosing Protocol: A Pre-Post Cohort Study"

_antibiotics, 2025, doi:10.3390/antibiotics14080775_

Round 1
Reviewer 1 Report
Comments and Suggestions for Authors
I congratulate the authors for their studies evaluating the clinical effect of implementing a structured standard operating procedure (SOP) for intravenous vancomycin treatment in orthopaedic patients.
You have specified severe renal dysfunction (stage 4 or higher chronic kidney disease according to KDIGO criteria) or patients requiring dialysis as exclusion criteria from the study. Do you prefer vancomycin treatment for these patient groups at your hospital?
I also believe that ethical approval is not necessary for the retrospective control group of the study. Still, it would have been appropriate to obtain ethical approval for the prospective section, which is the core of the study. Similarly, it would have been relevant to obtain consent from the patients included in the prospective section.
As you mentioned among your limitations, the most fundamental issue with the study is that the control group is retrospective, but even in this form, I believe your results are conclusive. Your detailed and honest sharing of the other limitations of your study is also highly valuable. Additionally, the section on future studies that should be conducted on this topic is inspiring.
Reviewer 2 Report
Comments and Suggestions for Authors
This paper is a single-center, pre-post cohort study evaluating the clinical impact of implementing a structured standard operating procedure (SOP) for intravenous vancomycin therapy in orthopedic inpatients. The intervention consisted of a standard operating procedure (SOP) for intravenous vancomycin therapy (56 patients prospectively included) after a Retrospectively analyzed Control group before implementation of SOP (n=58). Primary outcome was the incidence of vancomycin-associated acute kidney injury. Secondary outcomes included therapeutic trough level attainment, and infusion-related or ototoxic adverse events.
It is a very interesting and useful study to provide tools to improve the outcome of patients requiring vancomycin. SOP implementation needs to be clarified. Indeed, many tools are available for AMS. But if teams don't have the means to use them, then the tool loses its value, and so do the study results.
Comments on the text :
P.2 1 introduction
The rationale emphasizes the importance of MRSA in the orthopedic surgical site infections. The coagulase negative staphylococci burden must be exposed in chronic implant related infection especially since Staphylococcus epidermidis is the most common pathogen in this study isolated from patients receiving targeted therapy.
P.3 2.2. Patient selection:
Precise the number of excluded patients, particularly those receiving vancomycine and aminosid because they are a target for adverse events. A flow chart would be more informative.
Unacid : please use INN (international Non proprietary Name)
p.3 : 2.3. Intervention: Standard Operating Procedure
“In March 2024, a structured standard operating procedure (SOP) for intravenous vancomycin therapy was implemented at the Department of Orthopedic Surgery” : It would be interesting to have details about the implementation of the SOP. Indeed, it is important to find out whether it can be reproduced, particularly in terms of the human resources required.
P.5 3.1. Patient characteristics :
All informations given about the age of included patients are unnecessary (mean and median ge AND Figure 1). A table with patient characteristics in the two groups would be more informative than the figures 1 to 4.
There is a confusion between control group data and intervention group ones. We don’t understand why some data are givent only for intervention group (e.g. : Fig.2 and 3) and sometimes not precised (e.g.: fig.4)
- A table with patient characteristics in the two groups would be more informative
P.8 : 3.2. Loading dose
“all 23 patients (100.0%) in the intervention group were administered a weight-based loading dose in accordance with the standardized protocol” : why 23 patients represent 100% of intervention group patients?
P.8 : 3.2. Loading dose
It would be interesting to have informations about the time to onset of renal failure in relation to the start of vancomycin treatment to identify key moments of monitoring in the SOP.
Discussion :
The discussion is well conducted. The antibiotic stewardship is a very important point about the interest of this intervention in terms of improving patient outcomes (“reducing antibiotic-related harm” AND increasing efficacy) and not only “combating resistance”.
It would be interesting to talk about alternative treatment, as the daptomycin increasingly used, easier to monitor with a high bactericidal activity and about the possible limitations related to the cost and human resources required to implement the SOP.
P.9 : 4.1. Principal Findings and Clinical Relevance
“All patients in the post-implementation group received a weight-based loading dose” : idem than in the Loading dose paragraph p.8 : 56 patients or 23 patients?
